# An Interactive Protocol to Measure a Driver's Situational Awareness

**Abhijat Biswas***
Carnegie Mellon University
Pittsburgh, PA, USA
abhijat@cmu.edu

**Pranay Gupta***
Carnegie Mellon University
Pittsburgh, PA, USA
pranaygu@andrew.cmu.edu

**David Held**
Carnegie Mellon University
Pittsburgh, PA, USA
dheld@andrew.cmu.edu

**Henny Admoni**
Carnegie Mellon University
Pittsburgh, PA, USA
henny@cmu.edu

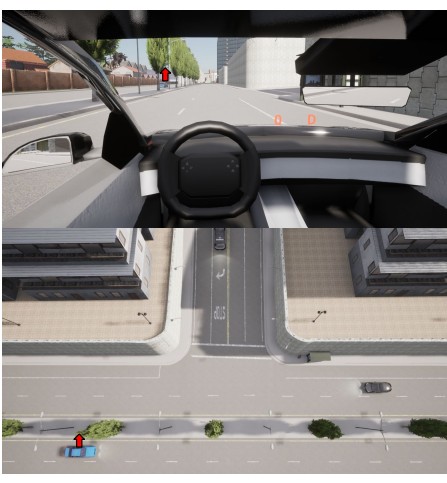 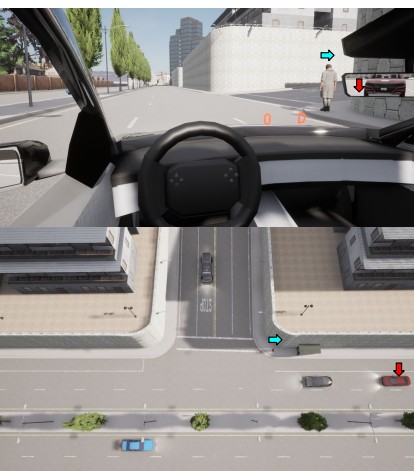 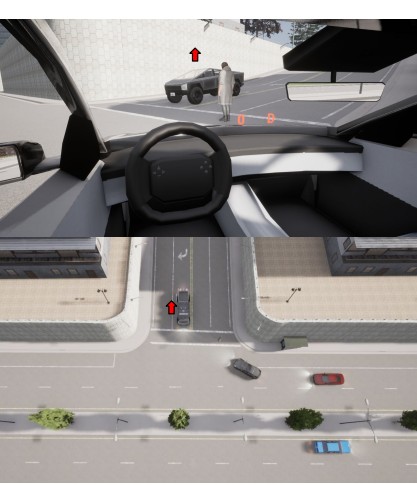

time

**Figure 1: Example sequence of driving scenes with per-object driver responses according to our proposed labeling method. The top row shows the scene from the driver view and bottom row shows the same scene via a birds-eye-view. Labels are shown as coloured arrows which correspond to buttons on the steering wheel. Responses are only required when the element is first seen by the driver. Red corresponds to vehicle labels and cyan to pedestrian labels.**

## ABSTRACT

Commonly used protocols for capturing the ground-truth situational awareness (SA) of drivers involve halting a simulation and querying the driver. SA data collected in this way is unsuitable for training models for predicting real-time SA since it is inherently intermittent and does not capture transitions of SA (e.g. from not aware to aware). We introduce an efficient VR based interactive protocol designed to capture a driver's ground-truth situational awareness (SA) in real time. Our protocol mitigates the aforementioned limitations of prior approaches, and allows capturing continuous object-level SA labels that are more suitable for downstream real-time SA prediction tasks. Our initial findings highlight its potential as a scalable solution for curating large scale driving datasets with ground-truth SA.

## CCS CONCEPTS

• **Human-centered computing** → **Systems and tools for interaction design**.

## KEYWORDS

Situational Awareness Grounding Methods, Assistive Driving, VR Driving Simulation

**ACM Reference Format:**
Abhijat Biswas*, Pranay Gupta*, David Held, and Henny Admoni. 2024. An Interactive Protocol to Measure a Driver's Situational Awareness. In *Proceedings of VAM-HRI'24*. ACM, New York, NY, USA, 5 pages.

Abhijat Biswas*, Pranay Gupta*, David Held, and Henny Admoni

# 1 INTRODUCTION

A driver's situational awareness (SA) encapsulates their knowledge of the unpredictably changing environment around them, hence guiding their choices and actions [9]. Insufficient SA contributes to a significant number of traffic accidents. A potential solution is to monitor a driver's SA and enhance it using intelligent alert systems [20]. However monitoring a driver's SA is not trivial. Naively counting the gazed-at objects is inefficient, as there is inherent ambiguity behind eye-gaze movements. Eye-gaze does not necessarily correspond to attention, as drivers might gaze at objects without gaining awareness due to effects like inattentional blindness [19]. Additionally, drivers can also gain cognizance of objects without gazing at objects through their peripheral vision [9]. Our goal is to address this ambiguity in eye-gaze movements to facilitate object level SA estimation. We do so in a data-centric fashion, and aim to model these effects using a large scale driving dataset with explicitly labeled ground-truth SA for all objects that enter the Field of View (FoV) of the driver.

Existing protocols for measuring the ground-truth SA of drivers collect data intermittently or sparsely. Recent methods [12, 23] adopt Situational Awareness Global Assessment Technique (SAGAT) [6] for measuring object level driver SA labels. This involved halting simulations and measuring SA through post-hoc queries about elements in the scene. However, these labels are intermittent, while they are accurate for when the simulation is paused they are not representative of the transition of the driver's attention from absent to present over each object. Modeling this transition is crucial for SA modulated safety alert systems for driving. Additionally, these protocols rely on the driver's ability to remember information about the scene elements, are also inefficient for large scale data collection.

In addition to these challenges, the labeling method should not affect the natural gaze behaviour of the drivers. Consider a labeling method that displays one of several icons over every scene element to be labeled which corresponds to icons on buttons that the user needs to press to label elements. This may cause the user to fixate on each scene element whereas otherwise they might use their peripheral vision to maintain awareness of that element —a clear distributional shift of gaze behavior. Models developed using this data would fail at test time as they might learn to associate fixations on or near scene elements as the only way to gain awareness of them.

Overcoming these challenges, we propose a novel interactive protocol for acquiring dense continuous driver SA labels for objects in their FoV. While driving a vehicle in a VR-based simulated driving environment [16], we instruct our annotators to perform a secondary awareness task. This awareness task requires users to indicate when they first become aware of an object by pressing a button on the steering wheel. In contrast to SAGAT and similar methods, we do not require the simulator to be paused and hence our method results in dense per-object labels which can collected continuously. Additionally, this protocol results in labels collected on the same timeline as driver gaze and simulated world events. Finally, it also does not affect the gaze behaviour of the driver as the secondary task only requires information that drivers must already acquire and process for safe driving. Refer 1 for example driving scenarios annotated with the driver responses for the awareness task.

---

*These authors contributed equally to this paper.

| SA Grounding Method | Capture Awareness Transition | Dense Object Labels | Doesn't Affect Natural Gaze Behaviour |
|---|---|---|---|
| SAGAT [8] | ✗ | ✓ | ✓ |
| DAZE [17] | ✓ | ✗ | ✗ |
| SPAM [4] | ✓ | ✗ | ✗ |
| Ours | ✓ | ✓ | ✓ |

**Table 1: Our protocol allows us to capture the transition in the driver's awareness of objects in the scene, allows labels for all objects in the scene without affecting the natural gaze behaviour of the driver.**

With this protocol, we conduct a user-study to curate our large-scale driving dataset. Our dataset contains sequences of driving events, the states of the ego vehicle, and other traffic elements, the eye-gaze of the driver and the SA labels for all the vehicles and pedestrians in the driver's FoV. To summarize the contributions of our work:

(1) We propose a interactive protocol for obtaining continuous and dense SA labels for on-road agents in a driving scene, without disrupting the driving task.
(2) We conduct a pilot user-study with our protocol which will eventually be used to curate a driving dataset with continuous per-object SA labels, on-road agent states, and driver eye gaze.

# 2 RELATED WORK

**Situational Awareness (SA) grounding methods:** The most popular situational awareness measurement technique is Situational Awareness Global Assessment Technique (or SAGAT) [6]. In SAGAT, the simulation is halted mid-task (e.g., simulated driving), and the driver is queried about the position, type and future status of elements within the scene to measure their SA. SAGAT was originally developed for flight interfaces, but has since been applied to driving tasks [8]. However, SAGAT provides intermittent labels, and hence does not capture the transition in the driver's awareness. The halting also limits the number of labels per drive that could be collected while maintaining the flow of simulation. Daze [17], and SPAM [4] mitigate halting issues by using real-time in situ questions. While they avoid pausing the simulation, they do not provide dense, per-object labels. Additionally, answering queries mid-task is disruptive and undesirably modifies gaze behavior. Table 1 compares our method to prior SA grounding methods, across their ability to generate continuous, dense object level SA without affecting the natural eye-gaze behaviour.

Physiological signals such as EEG [11], respiratory rate [18], and heart rate [13] have also been employed to measure SA. The most commonly used physiological technique was based on eye tracking. This included signals as blink rates, pupil dilation, but also behavioral characteristics such as fixation rates, dwell times, and saccade frequency to measure SA [22]. However, physiological methods are noisy, show small correlations with SA, and only provide an overall impression of SA rather than per-object SA.

**Driver's attention prediction datasets:** While we try to predict objects a particular driver is aware of, another line of work tries

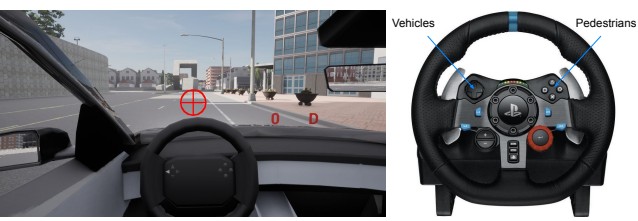

**Figure 2: The interface for collecting data using our protocol in VR simulator. Driver ego view during data collection (left) and hardware steering wheel (right) are shown. The hardware buttons used to solicit responses are mirrored on the steering wheel for user feedback while in the VR simulator. Each set of 4 buttons corresponds to a type of traffic element (Vehicles or Pedestrians). The left "Vehicles" button is currently being pressed and is hence lit up in the simulation. Driver eye gaze is also visualized (red reticle).**

to predict the regions most drivers would pay attention to [1, 14, 15, 21]. To do so, they leverage large scale datasets of front view RGB driving videos, where regions traversed by the driver's gaze are considered salient. While, these works assume that eye-gaze correlates with attention, in our dataset we explicitly label objects attended by the driver using our protocol. Dreyeve [15], is one of the largest datasets driver attention prediction dataset recorded in-car, while BDD-A [21] is the largest driver attention prediction dataset recorded in lab. More recent driver attention prediction datasets [1, 7] focus on accidents. Unlike these works, our dataset is recorded in-lab in a VR-based driving simulator [16] which closely replicates the driving environment. This allows us to enjoy the benefits of both in-car and in-lab curation strategies.

## 3 METHOD

In this section, we first explain our protocol for SA measurement, and then describe our user study design which will be eventually used for curating our novel dataset.

### 3.1 Protocol for SA measurement

We propose a novel protocol that allows drivers to indicate their object level SA while driving a vehicle in a VR-based driving simulation [16]. Along with the primary task of driving safely, according to traffic rules, we introduce a secondary awareness task. The awareness task requires users to indicate when they first become aware of an object by pressing a button on the steering wheel. We use two 4-way directional button palettes on the steering wheel, one to indicate vehicles and the other for pedestrians. Refer Fig 2, for an image of the interface. Typically, a user while driving in the VR simulation will press the button from the respective button palette when they first become aware of a vehicle or a pedestrian. They will press the directional buttons corresponding to the relative direction of the object when the driver first became aware of it.

This protocol, allows us to obtain dense continuous SA labels over objects in the scene and alleviates the need for halting the simulation as in SAGAT. Furthermore, the secondary task does not require any other information than is necessary for the primary task of safely driving, and hence does not affect the natural gaze behaviour. Finally,

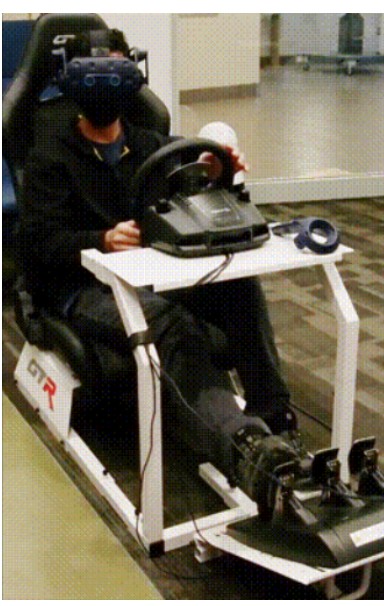

**Figure 3: Physical setup with a driver in a driving pose, driving the ego vehicle within the DReyeVR [16] simulator.**

our protocol results in labels collected on the same timeline as the driver's gaze and the simulated world events, unlike methods that use verbal natural responses which require manual annotation and may have timing inconsistencies. Our protocol can also be used to indicate per-object SA at all SA levels as defined in [5]. In the current form, labels indicate per-object situational awareness at SA levels 1 (perception —traffic elements must be perceived to be responded to) and 2 (comprehension —traffic elements are distinguished between vehicles and pedestrians). However, our protocol can be modified to incorporate level 3 SA by modifying the task objective to pressing directional buttons corresponding to the direction in which objects will travel in the next few seconds" but we consider only levels 1 and 2 in this work.

### 3.2 Pilot User Study

**Participants**: We pilot our protocol with users from within the Robotics Institute at CMU. We require our users to have a valid US issued driving license. Since, our physical rig associated with the VR simulator stays fixed while the user drives inside the simulator, it can sometimes induce nausea, hence it is necessary to identify whether our participants are fit to drive inside a VR simulator. Thus before collecting data we conduct trial rounds to verify whether users are fit to drive in the simulator. As a part of the trial rounds the users are instructed to drive a vehicle according to US driving laws, to a goal location in the simulator following in-world navigational signs.

**Instructions**: For the main rounds of the pilot study, the participants were instructed to do two tasks. For the primary task, they were asked to drive the ego vehicle in the simulator to a goal location by following navigational signs, by abiding the general US driving rules. The only difference was that right turns on red lights were not allowed. Along with this they were asked to do the secondary

awareness task. As a part of the secondary task we gave the following instruction: "When you first become aware of a vehicle or a pedestrian in your FoV, press a button from the respective 4-way button palette on the steering wheel. Within the 4 directional buttons choose the one that corresponds to the direction in your FOV the object appeared in." We conducted two trial rounds with each pilot participant, first to identify whether can drive in the simulator and the second to get acquainted with the system. In the first trial round we instructed them to only do the primary driving task. In the second trial round, we instructed them to do both the tasks. We conducted the main round after the two trials, where the users performed both the tasks, and we recorded their eye gaze, their awareness indications and other simulation data (For e.g., object states)

**Implementation Details** : We use DReyeVR [16] as the VR-driving simulator. DReyeVR extends the Carla [3] simulator to add virtual reality integration, a first-person maneuverable ego-vehicle, eye tracking support, and several immersion enhancements such as mirrors and sounds. Our physical setup includes a HTC Vive Pro Eye as the head-mounted VR device, which has built-in eye tracking, and an available eye tracking SDK. For our driving hardware we use a Logitech G29 wheel and pedals kit. Fig 3, shows the entire physical setup used for the study. For driving routes, we plan on using routes from the training maps of the Longest6 [2] benchmark, which is a commonly used benchmark in the autonomous driving community. Furthermore, we control the traffic in the simulation such that only a single vehicle or pedestrian enters the FoV of the driver from a single direction. If multiple objects enter the driver's FoV from the same direction at the same time, even if the user presses the corresponding directional buttons multiple times, we have no way of associating the button presses with the objects.

## 4   DISCUSSION

Our pilot study shows encouraging signs towards the development of our protocol as a reliable and scalable method for capturing dense, continuous object-level driver situational awareness data. However, there are certain limitations, which need to be addressed.

We have to rely on a VR based driving simulation to collect data. As mentioned before, driving in a VR simulation induces nausea (cyber sickness), which makes recruiting participants difficult. Not everyone is capable of driving a vehicle inside a VR simulator for a long time. Younger participants (such as university undergrauates) are generally less susceptible to cyber sickness but also have less-to-no driving experience. Thus, while our method is scalable, recruiting participants can be challenging. Furthermore, our protocol will be ineffective in collecting SA data in heavy traffic. Our protocol involves drivers performing a secondary task in addition to the primary task of safe driving. Hence, in cases when the cognitive load of driving increases the driver's performance on the secondary task will suffer, resulting in inaccurate SA labelling. Furthermore, in our current design we have 4 directional buttons for vehicles and pedestrians. Due to this we can only associate button clicks with a single vehicle or pedestrian in each direction. While some other input method like spoken-aloud natural language could resolve this issue, manual coding of varied responses can be challenging. Additionally, the button

presses are a crisp signal with a clear point indicating the transition from unaware to aware which is not true for natural language.

Drivers are not naturally deft at performing the secondary task. During our pilot we observed that participants mixed up between the two sets of buttons, i.e, they pressed a button for vehicles when they saw a pedestrian and vice-versa. This happened as the participants hand-eye coordination had not adjusted to perform the secondary task. However, we observed a decrease in such errors as the participants spent more time with our system. Finally, marking objects on turns was difficult —when the driver turns the steering wheels their hands move away from the buttons, and hence they have to recalibrate their hands to the buttons, once they complete the turn. Recalibrating hands in to the buttons on the physical steering wheel can be challenging since participants cannot see their hands while wearing the VR headset. To mitigate this, we readjusted the sensitivity of the steering wheel so that it was not necessary to turn the wheel to its extremes to accomplish turning in the simulator. However, these issues persist mainly due to the shift between driving in a real-world and driving within our simulator with a secondary task and can be mitigated by adding more trial rounds to increase familiarity with our system.

## 5   CONCLUSION

In this paper, we presented a novel protocol for capturing the dense and continuous object level ground-truth situational awareness of a driver. Prior methods were highly inefficient which made them unsuitable for curating large dataset driving dataset with continuous and dense object level ground-truth SA. Our protocol mitigates these adverse effects, and our pilot studies show encouraging results to position our protocol as a viable solution for obtaining continuous SA labels over objects in the scene at scale. Our pilots surfaced some limitations and we modified the study parameters to minimize their occurrence and effects. We plan to use our proposed protocol to curate a large scale dataset for training realtime driver SA estimation systems. In the future, we wish to develop a SA support driving system, that combines SA estimation methods with important object estimation methods [10] to alert drivers to important objects in the scene that they are unaware of.

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
