# OpenReview forum: "An Interactive Protocol to Measure a Driver’s Situational Awareness"
_humanrobotinteraction.org/HRI/2024/Workshop/VAM-HRI — VAM-HRI 2024 Oral_

### Official Review · Reviewer_eeaK · 2024-02-03
**review xap**

**Rating:** 7
**Confidence:** 5

**Review:**

In this paper, authors describe VR situation awareness (SA) user studies studies for driving scenarios. They focus on measuring gaze and drivers' eye-movements in the VR simulation. Their system enables drivers to obtain SA labels without disrupting the driving itself or stopping the simulation. The paper shows really interesting study and potential advancement in SA research. Especially a simple, yet effective way to label and monitor SA without interrupting the user is a great achievement.

Potential improvements:
1. "Refer 1 for example driving scenarios annotated with the driver responses for the awareness task." something is missing here, 3.1 "Refer Fig 2", refer to... In general, referring to figures/sections in this paper seems a bit off/rough.
2. Not 100% clear what is actually labeled. You elaborate on that in the methods section but until I reach  "Method" I am quite lost in what is being labeled. My first impression was that somehow you have to label the "gaze transition/situation awareness". Only after researching the method section, I understand that you have to label objects that you observed and then you measure SA by that. This is not clear in the introduction and clearly explaining your methodology there would improve the clarity of the paper.  Including something like paragraph 1 of sec 3.1. in the introduction would help me understand from the beginning what the paper is about. I think before submitting the full paper you should really work on the first two sections to be sure that you successfully communicate the motivation, research goal and methodology of your project.
3. You discuss the Pilot User study, but you do not analyze it in detail. While it is a workshop paper, it would be nice to get some insight into preliminary results.

---

### Official Review · Reviewer_nHW7 · 2024-02-03
**Review 2: Accept**

**Rating:** 7
**Confidence:** 5

**Review:**

Summary:
This paper introduces a situational awareness capturing protocol applied in a VR based driving simulator. Unlike prior methods, this protocol can capture awareness transitions, dense object labels, and does not affect natural gaze behavior, thus providing the potential for improved collected ground-truth SA datasets. This paper outlines the initial stages of the data collection process as well as current hurdles to their initial implementation.

Strengths:
- This paper discusses clear benefits of the proposed SA protocol over prior well known ones (SAGAT, DAZE, SPAM).
- By utilizing virtual reality, the paper provides a more realistic driving scenario for users. In turn, hopefully the data collection provides more accurate results as well.
- The paper provides a solid initial evaluation that resulted in important feedback that can be enacted upon in preparation for follow up work.

Weaknesses:
- It is currently unclear how this paper’s method might be utilized in the field of VAM-HRI. It would be helpful if the authors included this motivation within the introduction.
- As stated in the Discussion section, the four directional buttons are difficult to correlate with the vehicles and pedestrians, and can only map to a single vehicle or pedestrian in each direction. It could be pertinent to discuss whether natural language could be a better facilitator of object identification.

---

### Decision · Program_Chairs · 2024-02-10

Accept (Oral)